# Associations Between Lipid Profiles and Vitamin A and D Deficiencies Among Children and Adolescents in Zhejiang Province, China: A Cross-Sectional Study

**DOI:** 10.3390/nu17193159

**Published:** 2025-10-06

**Authors:** Yan Zou, Li-Chun Huang, Meng-Jie He, Dan Han, Dan-Ting Su, Pei-Wei Xu, Rong-Hua Zhang

**Affiliations:** Zhejiang Provincial Center for Disease Control and Prevention, Hangzhou 310051, China; zouyan0573@163.com (Y.Z.);

**Keywords:** lipid profiles, vitamin A deficiency, vitamin D deficiency, children, adolescents

## Abstract

Background: This study investigates the associations between lipid profiles, including triglycerides (TGs), total cholesterol (TC), high-density lipoprotein cholesterol (HDL-C), and low-density lipoprotein cholesterol (LDL-C), and vitamin A (VA) and vitamin D (VD) deficiencies among children and adolescents in Zhejiang Province, China. Methods: This cross-sectional study was conducted from 2022 to 2024 and included 9039 children and adolescents aged 6–17 years who participated in the provincial nutrition surveillance in Zhejiang Province. Blood samples were collected to measure the concentrations of TG, TC, HDL-C, LDL-C, VA and VD. Results: The prevalence of elevated TG, elevated TC, low HDL-C, and elevated LDL-C was 21.0%, 11.6%, 4.7% and 5.1% among children and adolescents aged 6–17 years, respectively. There were significant differences with respect to elevated TG and low HDL-C prevalence between sex (χ^2^ = 10.303 and 7.27, *p* = 0.006 and 0.026). There were significant differences with respect to elevated TC and low HDL-C prevalence between urban areas and rural areas (χ^2^ = 13.289 and 10.195, *p* = 0.001 and 0.006). There were significant differences with respect to elevated TG, elevated TC, low HDL-C, and elevated LDL-C prevalence among children with or without obesity/overweight (χ^2^ = 209.828, 58.54, 171.972, and 146.256, *p* < 0.001). There were significant differences with respect to elevated TC and low HDL-C prevalence among children with or without vitamin D deficiency/vitamin D insufficiency (χ^2^ = 33.37 and 56.848, *p* < 0.001). Ordinal regression analysis revealed that sex and obesity/overweight were associated with critical/abnormal TG (χ^2^ = 340.03, *p* < 0.001), sex, age group, location and obesity/overweight were associated with critical/abnormal TC (χ^2^ = 255.125, *p* < 0.001), age group, location, obesity/overweight and vitamin D deficiency/vitamin D insufficiency were associated with critical/abnormal HDL-C (χ^2^ = 458.527, *p* < 0.001), and age group, obesity/overweight were associated with critical/abnormal LDL-C (χ^2^ = 164.380, *p* < 0.001). Conclusions: Elevated TG, elevated TC, low HDL-C, and elevated LDL-C are prevalent in this population, with notable differences based on sex, urban vs. rural residence, and obesity/overweight status. Furthermore, vitamin D deficiency was linked to elevated TC and low HDL-C prevalence. Future interventions should focus on targeted public health strategies to mitigate these disparities and promote healthier lipid profiles in children and adolescents.

## 1. Introduction

Lipid abnormalities, including elevated triglycerides (TGs), total cholesterol (TC), low high-density lipoprotein cholesterol (HDL-C), and elevated low-density lipoprotein cholesterol (LDL-C), are significant risk factors for cardiovascular diseases (CVDs) and metabolic disorders [1,2]. These conditions often have their origins in childhood and adolescence, making early identification and intervention crucial for long-term health outcomes [3,4]. Previous studies have demonstrated that lipid profiles in children and adolescents can be influenced by various factors, including diet, lifestyle, and genetic predispositions.

Vitamin A and D deficiencies are also prevalent in many populations worldwide, particularly in children and adolescents [5,6]. Vitamin A is essential for maintaining normal vision, immune function, and growth [7], while vitamin D plays a critical role in calcium metabolism, bone health, and immune regulation [8]. Recent evidence suggests that vitamin deficiencies may also have an impact on lipid metabolism [9]. For instance, vitamin D deficiency has been associated with an increased risk of dyslipidemia and cardiovascular diseases in adults [10]. However, the relationship between vitamin A and D deficiencies and lipid profiles in children and adolescents remains less well understood.

In China, particularly in regions like Zhejiang Province, there is a growing concern about the prevalence of both lipid abnormalities and vitamin deficiencies among the pediatric population. Understanding the associations between lipid profiles and vitamin A and D deficiencies in this region could provide valuable insights for developing targeted public health interventions. This study aims to explore these associations among children and adolescents aged 6–17 years in Zhejiang Province, using data from a comprehensive provincial nutrition surveillance program.

## 2. Methods

### 2.1. Study Design and Participants

The cross-sectional study was conducted from 2022 to 2024 and included children and adolescents aged 6–17 years who participated in the provincial nutrition surveillance. The participants were recruited from 90 counties (cities and districts) across Zhejiang Province as part of the provincial nutrition surveillance program, which aims to assess the nutritional status of the pediatric population in the region. Blood samples were collected from the participants to measure the concentrations of triglycerides (TGs), total cholesterol (TC), high-density lipoprotein cholesterol (HDL-C), low-density lipoprotein cholesterol (LDL-C), vitamin A (VA), and vitamin D (VD), and no interventions were administered to the participants. A structured questionnaire from the Chinese Resident Nutrition and Health Survey (CNHS) were used [11].

### 2.2. Anthropometric Measurements

Anthropometric data were collected by trained research staff according to a standardized protocol. Height was measured without shoes to the nearest 0.2 cm using a portable stadiometer (TZG; Wuxi Weighing Instrument Factory Co., Ltd., Wuxi, China), and weight was measured without shoes and in light clothing to the nearest 0.1 kg on a calibrated beam scale (G&G TC-200k; Shanghai Taizhiheng Electronic Weighing Instrument Co., Ltd., Shanghai, China). Body mass index (BMI) was calculated as the weight in kilograms divided by the square of the height in meters.

### 2.3. Laboratory Assessment

Blood samples were collected after an overnight fast of at least 8 h. Participants were explicitly instructed to refrain from eating or drinking anything except water for the 8 h preceding the blood draw. This standard fasting protocol ensures that lipid measurements are not affected by recent food intake. Serum TG (triglyceride) levels were measured via a Roche Cobas 8000 analyser (Roche, Basel, Switzerland), whereas total cholesterol (TC) levels were measured via a Hitachi LST008AS analyser. HDL-C (high-density lipoprotein cholesterol) and LDL-C (low-density lipoprotein cholesterol) levels were measured using the direct method via the Hitachi LST008AS.

Abnormal lipid profiles were defined according to the Chinese Guidelines for Blood Lipid Management (2023) as follows [12]:For children aged 6–9 years, TG ≥ 1.1 mmol/LFor children aged 10–17 years, TG ≥ 1.5 mmol/LTC ≥ 5.2 mmol/LSLDL-C ≥ 3.4 mmol/LHDL-C ≤ 1.0 mmol/L.

Critical lipid profiles were defined according to the Chinese Guidelines for Blood Lipid Management (2023) as follows [12]:For children aged 6–9 years, TG ≥ 0.8 mmol/LFor children aged 10–17 years, TG ≥ 1.0 mmol/LTC ≥ 4.4 mmol/LLDL-C ≥ 2.8 mmol/LHDL-C ≤ 1.2 mmol/L

Vitamin D levels were measured via high-performance liquid chromatography-tandem mass spectrometry (HPLC-MS/MS) (AB SCIEX 5500MPX) as recommended by the Chinese national standard (WS/T677-2020) [13]. Vitamin D deficiency and insufficiency were assessed on the basis of the concentration of serum 25-hydroxyvitamin D (25-(OH)D) [13]. Specifically,
Vitamin D deficiency was defined as a serum 25-(OH)D level less than 12 μg/L.Vitamin D insufficiency was defined as a serum 25-(OH)D level less than 20 μg/L.

Vitamin A levels were measured via high-performance liquid chromatography (HPLC) (WS/T553-2017) [14]. Vitamin A deficiency and marginal deficiency were assessed on the basis of the concentration of serum retinol, according to the Chinese national standard [14]. Specifically,
Vitamin A deficiency was defined as a serum retinol concentration less than 0.2 μg/mL.Marginal vitamin A deficiency was defined as serum retinol levels less than 0.3 μg/mL but greater than or equal to 0.2 μg/mL.

### 2.4. Obesity Status

We defined being overweight and obese using the sex- and age-specific BMI (kg/m^2^) cut-off points [15]. Overweight and obese children were screened (Screening for Overweight and Obesity Among School-Age Children and Adolescents; issued by the State Health and Family Planning Commission of the People’s Republic of China), and the prevalence of being overweight and obese was calculated.

### 2.5. Ethical Considerations

All procedures performed in this study were in accordance with the ethical standards of the institutional research committee. Ethical approval was obtained from the Ethics Committee of the Zhejiang Provincial Center for Disease Control and Prevention (approval number: 2022-018-01), and informed consent was obtained from all individual participants included in the study.

### 2.6. Statistical Analysis

Median, 25% and 75% percentiles are presented for the continuous variables. Comparisons were performed using Mann–Whitney U nonparametric test for the continuous variables and chi-squared test for categorical variables. To further evaluate the differences in means for normally distributed continuous variables, Student’s *t*-tests were conducted. Additionally, multivariate analysis of variance (MANOVA) was performed to assess the overall differences in lipid profiles across different groups. Ordinal regression models were employed to explore the influencing factors of critical/abnormal TG, TC, HDL-C, and LDL-C. Data processing and statistical analyses were performed using SAS 9.2 software (SAS Institute, Cary, NC, USA). All tests were two-sided, and the level of significance was set at *p* < 0.05.

Power Calculation: The sample size for this study was determined based on the prevalence of dyslipidemia in previous studies. Assuming a prevalence of 20%, a power of 80%, and a significance level of 0.05, the required sample size was calculated to be 1537 participants per group. Given that our study included both urban and rural children and adolescents, and considering the need for adequate representation from both groups, the total sample size was adjusted accordingly. The final sample size of 9039 participants, accumulated over three years of monitoring (2022–2024), ensures sufficient statistical power to detect significant differences in lipid profiles across various subgroups. This comprehensive sample size allows for robust analysis and meaningful conclusions regarding the prevalence and determinants of dyslipidemia in the pediatric population of Zhejiang Province.

## 3. Results

### 3.1. Study Population

The study included 9039 children and adolescents, and the response rate was 94.2%. Table 1 and Table 2 shows serum lipid concentrations by location and sex among children and adolescents in Zhejiang, 2022–2024. Significant differences in serum total cholesterol (TC), high-density lipoprotein cholesterol (HDL-C), and low-density lipoprotein cholesterol (LDL-C) levels were observed between urban and rural children aged 6–17 years. Specifically, the mean serum TC level in urban children was 4.24 mg/dL, compared to 4.19 mg/dL in rural children (*p* = 0.000). Similarly, the mean HDL-C level was 1.53 mg/dL in urban children and 1.49 mg/dL in rural children (*p* = 0.000). For LDL-C, the mean levels were 2.3 mg/dL in urban children and 2.26 mg/dL in rural children (*p* = 0.000). These findings indicate that there are notable disparities in lipid profiles between urban and rural children within this age group. Significant differences in serum TG levels were observed between sex in children aged 6–17 years. The mean serum TG level in boys was 0.83 mg/dL, while in girls it was 0.9 mg/dL (*p* = 0.000). The mean serum TC level in boys was 4.19 mg/dL, while in girls it was 4.23 mg/dL (*p* = 0.046). This indicates that there are notable sex disparities in serum TG levels among children in this age group.

### 3.2. Abnormal Rates of Serum Lipids

Table 3 and Table 4 shows the critical and abnormal rates of serum lipids among children and adolescents with and without overweight/obesity during 2022–2024. There were significant differences in the prevalence of dyslipidemia and the levels of serum triglycerides (TGs), total cholesterol (TC), high-density lipoprotein cholesterol (HDL-C), and low-density lipoprotein cholesterol (LDL-C) between children and adolescents with and without overweight/obesity (χ^2^ = 209.808, 58.54, 171.972, 146.256, respectively; *p* < 0.001 for all).

Specifically, the critical rate of TG was 25.3%/27.8% in children and adolescents with overweight/obesity compared to 24.4% in those without (*p* = 0.000). The abnormal rate of TC was 28.7%/32.7% in the overweight/obesity group and 28.0% in the non-overweight/obesity group (*p* = 0.000). For HDL-C, the critical rate was 15%/16.2% in the overweight/obesity group and10.6% in the non-overweight/obesity group (*p* = 0.000). Lastly, the abnormal rate of LDL-C was 18.9%/20.1% in the overweight/obesity group and 12.7% in the non-overweight/obesity group (*p* = 0.000).

### 3.3. Vitamin A and Vitamin D Status and Blood Lipid Profiles

The associations between vitamin A status and blood lipid profiles among children and adolescents aged 6–17 years are summarized in Table 5 and Table 6. The critical rates of TG were 25.5% for normal vitamin A status, 15.6% for marginal deficiency, and 14.3% for deficiency. The abnormal rates were 21.8%, 7.7%, and 14.3%, respectively. These differences were statistically significant (χ^2^ = 110.494, *p* < 0.001). The critical rates of TC were 29.1% for normal vitamin A status, 20.1% for marginal deficiency, and 18.8% for deficiency. The abnormal rates were 11.9%, 5.6%, and 11.6%, respectively. These differences were statistically significant (χ^2^ = 54.968, *p* < 0.001). The critical rates of HDL-C were 11.7% for normal vitamin A status, 15.1% for marginal deficiency, and 12.5% for deficiency. The abnormal rates were 4.7%, 4.3%, and 12.5%, respectively. These differences were statistically significant (χ^2^ = 10.322, *p* = 0.035). The critical rates of LDL-C were 14.8% for normal vitamin A status, 7.6% for marginal deficiency, and 9.4% for deficiency. The abnormal rates were 5.2%, 3.0%, and 9.4%, respectively. These differences were statistically significant (χ^2^ = 29.985, *p* < 0.001).

The associations between vitamin D status and blood lipid profiles among children and adolescents aged 6–17 years are summarized in Table 7 and Table 8.

No significant differences were observed in the critical and abnormal rates of TG across different vitamin D status groups (χ^2^ = 6.155, *p* = 0.188). Specifically, the critical rates of TG were 24.4% for those with normal vitamin D status, 25.4% for those with vitamin D insufficiency, and 26.1% for those with vitamin D deficiency. The abnormal rates were 20.4%, 21.9%, and 20.6%, respectively. Significant differences were observed in the critical and abnormal rates of TC across different vitamin D status groups (χ^2^ = 33.37, *p* < 0.001). The critical rates of TC were 30.0% for those with normal vitamin D status, 27.1% for those with vitamin D insufficiency, and 26.6% for those with vitamin D deficiency. The abnormal rates were 12.7%, 10.6%, and 8.9%, respectively. Significant differences were observed in the critical and abnormal rates of HDL-C across different vitamin D status groups (χ^2^ = 56.848, *p* < 0.001). The critical rates of HDL-C were 10.3% for those with normal vitamin D status, 13.0% for those with vitamin D insufficiency, and 16.3% for those with vitamin D deficiency. The abnormal rates were 3.8%, 5.4%, and 6.9%, respectively. No significant differences were observed in the critical and abnormal rates of LDL-C across different vitamin D status groups (χ^2^ = 2.092, *p* = 0.719). The critical rates of LDL-C were 14.4% for those with normal vitamin D status, 14.3% for those with vitamin D insufficiency, and 14.3% for those with vitamin D deficiency. The abnormal rates were 5.4%, 4.8%, and 4.6%, respectively.

### 3.4. Ordinal Regression Analysis

The results of the ordinal regression analysis are shown in Table 9, Table 10, Table 11 and Table 12. The analysis revealed that age group (OR = 0.89, 95% CI = 0.80–0.99, *p* = 0.027) and weight status were significant determinants of LDL levels in children and adolescents. Specifically, both normal weight status (OR = 0.48, 95% CI = 0.41–0.55, *p* = 0.000) and overweight status (OR = 0.77, 95% CI = 0.64–0.92, *p* = 0.004) were associated with lower LDL levels compared to the reference group.

Additionally, the analysis also examined the factors associated with HDL, TG, and TC. For HDL levels in children and adolescents, age group (OR = 2.44, 95% CI = 2.17–2.74, *p* = 0.000) and location (OR = 1.17, 95% CI = 1.05–1.32, *p* = 0.006) were significant determinants. Weight status, including normal weight (OR = 0.39, 95% CI = 0.33–0.45, *p* = 0.000) and overweight (OR = 0.75, 95% CI = 0.62–0.91, *p* = 0.003), was associated with lower HDL levels compared to the reference group. Similarly, vitamin D status, including normal (OR = 0.68, 95% CI = 0.56–0.82, *p* = 0.000) and insufficient (OR = 0.81, 95% CI = 0.67–0.98, *p* = 0.029), was associated with lower HDL levels compared to the reference group.

For TG levels in children and adolescents, sex was significant (OR = 1.21, 95% CI = 1.12–1.31, *p* = 0.000). Weight status, including normal weight (OR = 0.44, 95% CI = 0.39–0.50, *p* = 0.000) and overweight (OR = 0.72, 95% CI = 0.62–0.84, *p* = 0.000), was associated with lower TG levels compared to the reference group.

For TC levels in children and adolescents, sex (OR = 1.12, 95% CI = 1.03–1.22, *p* = 0.005), age group (OR = 0.65, 95% CI = 0.59–0.70, *p* = 0.000) and location (OR = 0.91, 95% CI = 0.84–0.99, *p* = 0.026) were significant determinants. Weight status, including normal weight (OR = 0.68, 95% CI = 0.6–0.77, *p* = 0.000) and overweight (OR = 0.81, 95% CI = 0.69–0.94, *p* = 0.007), was associated with lower HDL levels compared to the reference group.

## 4. Discussion

This study provides valuable insights into the prevalence and influencing factors of abnormal lipid profiles among children and adolescents in Zhejiang Province, China. The findings reveal a significant burden of dyslipidemia, with notable differences based on sex, urban–rural residence, and obesity/overweight status. These results highlight the need for targeted public health interventions to address these disparities and promote healthier lipid profiles in children and adolescents.

### 4.1. Prevalence of Dyslipidemia

The prevalence of elevated TG (21%), elevated TC (11.6%), low HDL-C (4.7%), and elevated LDL-C (5.1%) among children and adolescents aged 6–17 years in Zhejiang Province is concerning. These rates are consistent with previous studies in China and other countries, indicating a global trend of increasing dyslipidemia in pediatric populations [16]. The high prevalence of dyslipidemia underscores the importance of early intervention to prevent long-term cardiovascular risks.

### 4.2. Sex Differences

Our study revealed significant sex and urban–rural disparities in serum lipid levels. Boys had higher mean serum TG levels, while girls had higher mean serum TC levels. Urban children had higher mean serum TC, HDL-C, and LDL-C levels compared to their rural counterparts. These findings are in line with previous research that has identified sex differences in serum lipid levels [17]. Although our dataset did not include Tanner stages to further investigate the interaction between sex and pubertal development, we recognize the importance of this factor. Future studies should consider incorporating Tanner stages to elucidate the relationship between gender, puberty, and lipid profiles. The observed gender differences may be attributed to variations in lifestyle, dietary habits, and genetic predispositions. Future interventions should account for these differences to develop effective public health strategies.

### 4.3. Urban–Rural Disparities

Obesity and overweight status were significantly associated with elevated TG, TC, low HDL-C, and elevated LDL-C. Children and adolescents with obesity or overweight had higher prevalence rates of dyslipidemia compared to their normal-weight counterparts. These findings suggest that urbanization may be associated with unfavorable changes in lipid profiles, possibly due to differences in diet, physical activity levels, and other lifestyle factors between urban and rural areas [18,19]. Public health initiatives should focus on addressing these disparities to promote equitable health outcomes for children and adolescents in both urban and rural settings.

### 4.4. Obesity and Overweight Status

The global obesity rate in children is rapidly increasing and has increased more than 8-fold in the past 40 years, and a healthy nutrition transition that enhances access to nutritious foods is needed [20]. Obesity and overweight status were significantly associated with elevated TG, TC, low HDL-C, and elevated LDL-C. Children and adolescents with obesity or overweight had higher prevalence rates of dyslipidemia compared to their normal-weight counterparts. This finding is consistent with previous studies demonstrating that obesity is a significant risk factor for dyslipidemia [21]. The metabolic interlink between hypertriglyceridemia and low HDL-C levels is strongly associated with cardiovascular disease (CVD) in both adults and children [1]. Our analysis also found that the prevalence of obesity and overweight was significantly different between boys and girls and this may be related to differences in growth patterns and lifestyle habits during the peripubertal period. Obesity and overweight can lead to insulin resistance, which in turn affects lipid metabolism. Insulin resistance often results in increased hepatic production of very low-density lipoprotein (VLDL), leading to elevated TG levels. Additionally, it impairs the clearance of TG-rich lipoproteins, further exacerbating hypertriglyceridemia [22]. Therefore, interventions targeting weight management and healthy lifestyle promotion are crucial to mitigate the risk of dyslipidemia in this population. Public health initiatives should focus on promoting a balanced diet rich in fruits, vegetables, whole grains, and healthy fats, while reducing the intake of saturated and trans fats. Regular physical activity should be encouraged to improve overall metabolic health and reduce the risk of obesity-related complications [23].

### 4.5. Vitamin A Status

In our study, we observed significant associations between vitamin A status and lipid profiles in the univariate analysis. Vitamin A deficiency was associated with higher critical and abnormal rates of TC and LDL-C. These findings suggest that maintaining adequate levels of vitamin A may be important for healthy lipid profiles in children and adolescents. Vitamin A deficiency has been associated with alterations in lipid profiles, possibly through its effects on liver function and lipid transport mechanisms [24]. However, these associations did not reach statistical significance in the ordinal regression analysis, possibly due to the adjustment for multiple confounding variables, small sample sizes in certain subgroups, and the complex interplay of other unmeasured factors.

### 4.6. Vitamin D Status

In contrast, vitamin D deficiency was significantly associated with elevated TC and low HDL-C in the ordinal regression analysis. This finding underscores the importance of vitamin D in lipid metabolism. Vitamin D plays a critical role in calcium metabolism, bone health, and immune regulation. Recent evidence suggests that vitamin D plays a role in regulating lipid metabolism, and its deficiency may contribute to adverse lipid profiles [25]. The mechanisms underlying this association are not fully understood, but vitamin D is known to influence lipid metabolism through its effects on the parathyroid hormone (PTH) and calcium levels, which in turn affect lipid synthesis and transport [26].

### 4.7. Mechanisms and Future Research

The mechanisms underlying the associations between vitamin A and D deficiencies and lipid profiles are not fully understood. Vitamin A is essential for the proper functioning of the retinoid X receptor (RXR), which is involved in the regulation of lipid homeostasis [27]. Deficiency in vitamin A may impair the activity of RXR, leading to dysregulation of lipid synthesis and transport. Vitamin D also plays a role in lipid metabolism, and its deficiency may contribute to adverse lipid profiles. Future studies should explore the underlying mechanisms and the potential benefits of vitamin supplementation in improving lipid profiles in children and adolescents.

### 4.8. Strengths and Limitations

#### 4.8.1. Strengths

Our study included three independent cross-sectional surveys conducted in 2022, 2023, and 2024, each capturing a representative sample of children and adolescents in Zhejiang Province. This approach allows us to assess the prevalence and associations of lipid profiles and vitamin deficiencies at different time points, enhancing the robustness of our findings and providing a comprehensive understanding of the current health landscape in the province. While cross-sectional studies cannot establish causality, they are well-suited for identifying prevalence and associations within a population at a specific point in time. The study employed ordinal regression analysis to explore the factors associated with critical/abnormal lipid profiles. This statistical approach allows for the simultaneous consideration of multiple variables, providing a comprehensive understanding of the factors influencing lipid profiles in children and adolescents. The findings highlight significant disparities in lipid profiles based on sex, urban–rural residence, and obesity/overweight status. These insights are crucial for developing targeted public health interventions aimed at reducing the prevalence of dyslipidemia and improving overall health outcomes in children and adolescents.

#### 4.8.2. Limitations

The study’s cross-sectional nature limits the ability to establish causality between the variables examined. While associations between lipid profiles and vitamin deficiencies were identified, further longitudinal studies are needed to determine the temporal relationships and causal pathways. The study was conducted in Zhejiang Province, which may not be representative of other regions in China or other countries. Regional variations in diet, lifestyle, and environmental factors can influence lipid profiles and vitamin status. Therefore, the findings may not be generalizable to other populations. Additionally, we did not collect data on family history of metabolic diseases or detailed dietary habits, such as eating out, picky eating, or food preferences. These factors may play a role in lipid profiles and vitamin status. In future studies, we plan to include these variables to better understand their impact. Although we analyzed the relationship between individual physical activity levels and lipid indicators and found no association, it is still important to consider the potential influence of family history and diet in future research.

## 5. Conclusions

In conclusion, this study highlights the significant prevalence of dyslipidemia and the disparities based on sex, urban–rural residence, and obesity/overweight status among children and adolescents in Zhejiang Province. Vitamin D deficiency was also found to be associated with adverse lipid profiles. These findings emphasize the need for targeted public health interventions to address these disparities and promote healthier lipid profiles in children and adolescents. Future research should focus on exploring the underlying mechanisms of these associations and developing effective interventions to improve lipid profiles and overall health in this population.

## Figures and Tables

**Table 1 nutrients-17-03159-t001:** Serum lipid concentrations by location and sex among children and adolescents in Zhejiang, 2022–2024.

Age	Blood Lipids (mmol/L)	Total			Urban Area			Rural Area			Male			Female		
		Median	25%	75%	Median	25%	75%	Median	25%	75%	Median	25%	75%	Median	25%	75%
6–17	TG	0.86	0.62	1.25	0.86	0.61	1.23	0.87	0.62	1.25	0.83	0.59	1.23	0.9	0.65	1.26
(N = 9039)	TC	4.21	3.77	4.72	4.24	3.8	4.75	4.19	3.74	4.71	4.19	3.74	4.73	4.23	3.79	4.72
	HDL-C	1.51	1.28	1.77	1.53	1.3	1.79	1.49	1.27	1.75	1.51	1.27	1.78	1.51	1.3	1.75
	LDL-C	2.28	1.92	2.68	2.3	1.96	2.69	2.26	1.91	2.68	2.27	1.92	2.69	2.29	1.93	2.68
6–11	TG	0.80	0.58	1.16	0.8	0.57	1.15	0.80	0.580	1.160	0.76	0.55	1.10	0.86	0.62	1.22
(N = 5375)	TC	4.31	3.86	4.83	4.35	3.88	4.83	4.29	3.85	4.83	4.32	3.86	4.84	4.31	3.85	4.82
	HDL-C	1.59	1.36	1.85	1.61	1.39	1.86	1.58	1.34	1.85	1.62	1.39	1.87	1.57	1.34	1.83
	LDL-C	2.31	1.97	2.71	2.32	1.99	2.72	2.30	1.96	2.71	2.31	1.97	2.72	2.31	1.97	2.70
12–17	TG	0.80	0.58	1.16	0.98	0.68	1.41	1.45	1.24	1.67	0.97	0.68	1.35	0.97	0.69	1.36
(N = 3664)	TC	4.11	3.65	4.61	4.06	3.61	4.58	4.17	3.71	4.65	4.13	3.67	4.6	4.1	3.65	4.62
	HDL-C	1.41	1.21	1.63	1.36	1.18	1.59	1.45	1.24	1.67	1.40	1.20	1.66	1.41	1.21	1.62
	LDL-C	2.26	1.91	2.66	2.24	1.91	2.66	2.29	1.91	2.67	2.26	1.92	2.65	2.26	1.90	2.67

**Table 2 nutrients-17-03159-t002:** The statistical evaluations of Serum lipid concentrations by location and sex.

Age	Blood Lipids (mmol/L)	Urban Area	Rural Area	Male	Female
		Z	*p*	Z	*p*
6–17	TG	1.088	0.277	5.708	0.000
(N = 9039)	TC	−3.215	0.001	1.997	0.046
	HDL-C	−4.938	0.000	0.267	0.790
	LDL-C	−2.917	0.004	0.348	0.728
6–11	TG	0.422	0.673	8.018	0.000
(N = 5375)	TC	−2.807	0.005	−0.742	0.458
	HDL-C	−4.170	0.000	−5.325	0.000
	LDL-C	−2.125	0.034	−0.393	0.694
12–17	TG	0.796	0.426	−1.518	0.129
(N = 3664)	TC	−1.190	0.234	4.376	0.000
	HDL-C	−1.290	0.197	8.028	0.000
	LDL-C	−1.803	0.071	1.103	0.270

**Table 3 nutrients-17-03159-t003:** Critical and abnormal rates of serum lipids among children and adolescents with normal weight, overweight, and obesity, stratified by age group.

Age	Blood Lipids (mmol/L)	Total (N = 9039)		Normal Weight (N = 6767)		Overweight (N = 1213)		Obesity (N = 1059)		χ^2^	*p*
		Critical Rate	Abnormal Rate	Critical Rate	Abnormal Rate	Critical Rate	Abnormal Rate	Critical Rate	Abnormal Rate		
6–17	TG	2256 (24.9%)	1900 (21.0%)	1655 (24.4%)	1211 (17.9%)	307 (25.3%)	337 (27.8%)	294 (27.8%)	352 (33.2%)	209.828	0.000
(N = 9039)	TC	2587 (28.6%)	847 (9.4%)	1895 (28.0%)	711 (10.5%)	349 (28.7%)	165 (13.6%)	346 (32.7%)	172 (16.3%)	58.54	0.000
	HDL-C	1065 (11.9%)	421 (4.7%)	717 (10.6%)	223 (3.3%)	181 (15.0%)	100 (8.2%)	170 (16.2%)	99 (9.5%)	171.972	0.000
	LDL-C	1302 (14.4%)	461 (5.1%)	859 (12.7%)	284 (4.2%)	231 (18.9%)	74 (6.1%)	213 (20.1%)	103 (9.8%)	146.256	0.000
6–11	TG	1204 (22.4%)	1204 (22.4%)	852 (21.8%)	774 (19.8%)	162 (22.7%)	188 (26.4%)	190 (25.3%)	240 (31.9%)	80.941	0.000
(N = 5375)	TC	1730 (32.2%)	488 (12.8%)	1243 (31.8%)	461 (11.8%)	222 (31.1%)	107 (15.0%)	265 (35.3%)	118 (15.7%)	20.437	0.000
	HDL-C	435 (8.1%)	183 (3.4%)	297 (7.6%)	94 (2.4%)	65 (9.1%)	35 (4.9%)	77 (10.2%)	50 (6.7%)	52.004	0.000
	LDL-C	822 (15.3%)	274 (5.1%)	524 (13.4%)	176 (4.5%)	149 (20.9%)	38 (5.3%)	154 (20.5%)	63 (8.4%)	71.826	0.000
12–17	TG	1052 (28.7%)	696 (19.0%)	803 (28.1%)	435 (5.2%)	145 (29.0%)	149 (29.8%)	104 (33.9%)	112 (36.5%)	157.281	0.000
(N = 3664)	TC	857 (23.4%)	359 (9.8%)	652 (22.8%)	249 (8.7%)	127 (25.3%)	58 (11.5%)	81 (26.4%)	54 (17.6%)	36.533	0.000
	HDL-C	630 (17.2%)	238 (6.5%)	420 (14.7%)	129 (4.5%)	116 (23.1%)	65 (12.0%)	93 (30.4%)	49 (16.1%)	184.386	0.000
	LDL-C	473 (12.9%)	187 (5.1%)	334 (11.7%)	109 (3.8%)	82 (16.3%)	36 (7.2%)	59 (19.1%)	40 (13.1%)	86.438	0.000

**Table 4 nutrients-17-03159-t004:** Critical and abnormal rates of serum lipids among children and adolescents with normal weight, overweight, and obesity, stratified by gender.

Age	Blood Lipids (mmol/L)	Total (N = 9039)		Normal Weight (N = 6767)		Overweight (N = 1213)		Obesity (N = 1059)		χ^2^	*p*
		Critical Rate	Abnormal Rate	Critical Rate	Abnormal Rate	Critical Rate	Abnormal Rate	Critical Rate	Abnormal Rate		
6–17	TG	2256 (24.9%)	1900 (21.0%)	1651 (24.4%)	1211 (17.9%)	307 (25.3%)	337 (27.8%)	294 (27.8%)	352 (33.2%)	209.828	0.000
(N = 9039)	TC	2587 (28.6%)	849 (9.4%)	1895 (28.0%)	711 (10.5%)	349 (28.7%)	165 (13.6%)	346 (32.7%)	172 (16.3%)	58.54	0.000
	HDL-C	1065 (11.9%)	421 (4.7%)	717 (10.6%)	223 (3.3%)	181 (15.0%)	100 (8.2%)	170 (16.2%)	99 (9.5%)	171.972	0.000
	LDL-C	1302 (14.4%)	461 (5.1%)	859 (12.7%)	284 (4.2%)	231 (18.9%)	74 (6.1%)	213 (20.1%)	103 (9.8%)	146.256	0.000
Male	TG	1129 (24.1%)	947 (20.2%)	750 (22.9%)	532 (16.3%)	194 (26.0%)	196 (26.3%)	185 (27.7%)	221 (33.1%)	156.730	0.000
(N = 4687)	TC	1298(27.7%)	548(11.7%)	875(26.7%)	320(9.8%)	210(28.2%)	115(15.3%)	218(32.6%)	115(17.3%)	66.805	0.000
	HDL-C	567(12.1%)	244(5.2%)	343(10.5%)	111(3.5%)	116(15.5%)	66(8.3%)	105(16%)	65(9.7%)	104.243	0.000
	LDL-C	680(14.5%)	248(5.3%)	384(11.6%)	127(3.9%)	159(21.2%)	47(6.3%)	139(20.9%)	73(10.9%)	149.163	0.000
Female	TG	1126(25.9%)	953(21.8%)	901(25.9%)	679(19.4%)	113(24.2%)	141(30.2%)	109(27.9%)	131(33.5%)	71.811	0.000
(N = 4352)	TC	1289(29.6%)	301(6.9%)	1020(29.2%)	391(11.2%)	139(29.5%)	50(10.6%)	128(32.8%)	57(14.6%)	8.529	0.000
	HDL-C	498(11.6%)	177(4.1%)	374(10.7%)	112(3.2%)	65(14.1%)	34(7.2%)	65(16.5%)	34(8.6%)	60.993	0.000
	LDL-C	622(14.3%)	213(4.9%)	475(13.6%)	157(4.5%)	72(15.3%)	27(5.8%)	74(18.8%)	30(7.7%)	18.951	0.000

**Table 5 nutrients-17-03159-t005:** Critical and abnormal rates of serum lipids among children and adolescents with marginal vitamin A deficiency and vitamin A deficiency, stratified by age group.

Age	Blood Lipids (mmol/L)	Normal (N = 8542)		Marginal Vitamin A Deficiency (N = 469)		Vitamin A Deficiency (N = 28)		χ^2^	*p*
		Critical Rate	Abnormal Rate	Critical Rate	Abnormal Rate	Critical Rate	Abnormal Rate		
6–17	TG	2178 (25.5%)	1859 (21.8%)	73 (15.6%)	36 (7.7%)	4 (14.3%)	4 (14.3%)	110.494	0.000
(N = 9039)	TC	2627 (29.1%)	1077 (11.9%)	94 (20.1%)	26 (5.6%)	5 (18.8%)	3 (11.6%)	54.968	0.000
	HDL-C	1051 (11.7%)	420 (4.7%)	71 (15.1%)	20 (4.3%)	4 (12.5%)	4 (12.5%)	10.322	0.035
	LDL-C	1332 (14.8%)	469 (5.2%)	36 (7.6%)	14 (3.0%)	3 (9.4%)	3 (9.4%)	29.985	0.000
6–11	TG	1147 (22.9%)	1175 (23.4%)	54 (15.8%)	25 (7.3%)	3 (16.7%)	3 (16.7%)	75.875	0.000
(N = 5375)	TC	1735 (33.1%)	696(12.9%)	67 (20.6%)	20 (6.1%)	3 (18.2%)	3 (9.1%)	61.872	0.000
	HDL-C	401 (7.7%)	173 (3.3%)	46 (14.2%)	11 (3.6%)	3 (13.6%)	3 (13.6%)	29.672	0.000
	LDL-C	833 (15.9%)	275 (5.3%)	28 (8.4%)	9 (2.8%)	1 (4.5%)	3 (13.6%)	27.353	0.000
12–17	TG	1031 (29.2%)	684 (19.4%)	19 (14.8%)	11 (8.6%)	1 (10.0%)	1 (10.0%)	34.625	0.000
(N = 3664)	TC	892 (23.6%)	381 (10.1%)	27 (19.0%)	6 (4.2%)	2 (20.0%)	0 (0.0%)	9.528	0.049
	HDL-C	650 (17.2%)	247 (6.5%)	25 (17.6%)	9 (6.3%)	1 (10.0%)	1 (10.0%)	0.554	0.968
	LDL-C	499 (13.2%)	194 (5.1%)	8 (5.6%)	5 (3.5%)	2 (20.0%)	0 (0.0%)	11.132	0.025

**Table 6 nutrients-17-03159-t006:** Critical and abnormal rates of serum lipids among children and adolescents with marginal vitamin A deficiency and vitamin A deficiency, stratified by gender.

Age	Blood Lipids (mmol/L)	Normal (N = 8542)		Marginal Vitamin A Deficiency (N = 469)		Vitamin A Deficiency (N = 28)		χ^2^	*p*
		Critical Rate	Abnormal Rate	Critical Rate	Abnormal Rate	Critical Rate	Abnormal Rate		
6–17	TG	2178 (25.5%)	1859 (21.8%)	73 (15.6%)	36 (7.7%)	4 (14.3%)	4 (14.3%)	110.494	0.000
(N = 9039)	TC	2627 (29.1%)	1077 (11.9%)	94 (20.1%)	26 (5.6%)	5 (18.8%)	3 (11.6%)	54.968	0.000
	HDL-C	1051 (11.7%)	420 (4.7%)	71 (15.1%)	20 (4.3%)	4 (12.5%)	4 (12.5%)	10.322	0.035
	LDL-C	1332 (14.8%)	469 (5.2%)	36 (7.6%)	14 (3.0%)	3 (9.4%)	3 (9.4%)	29.985	0.000
Male	TG	1093 (24.6%)	937 (21.1%)	33 (14.1%)	11 (4.7%)	3 (33.3%)	1 (11.1%)	69.168	0.000
(N = 4687)	TC	1262 (28.4%)	533 (12%)	37 (15.6%)	14 (6.2%)	1 (10%)	1 (10%)	39.573	0.000
	HDL-C	533 (12%)	231 (5.2%)	32 (13.8%)	13 (5.5%)	1 (10%)	1 (10%)	1.383	0.847
	LDL-C	662 (14.9%)	239 (5.4%)	15 (6.5%)	8 (3.3%)	1 (10%)	1 (10%)	22.108	0.000
Female	TG	1085 (26.5%)	922 (22.5%)	40 (17.0%)	25 (10.6%)	1 (5.3%)	3 (15.8%)	50.779	0.000
(N = 4352)	TC	1225 (29.9%)	488 (11.9%)	59 (24.9%)	12 (5.0%)	4 (22.7%)	2 (4.5%)	20.225	0.000
	HDL-C	463 (11.3%)	168 (4.1%)	39 (16.5%)	7 (3.1%)	3 (13.6%)	3 (13.6%)	9.811	0.044
	LDL-C	598 (14.6%)	205 (5%)	21 (8.8%)	6 (2.7%)	2 (9.1%)	2 (9.1%)	13.006	0.011

**Table 7 nutrients-17-03159-t007:** Critical and abnormal rates of serum lipids among children and adolescents with vitamin D insufficiency and deficiency, stratified by age group.

Age	Blood Lipids (mmol/L)	Normal (N = 4798)		Insufficiency (N = 3479)		Deficiency (N = 762)		χ^2^	*p*
		Critical Rate	Abnormal Rate	Critical Rate	Abnormal Rate	Critical Rate	Abnormal Rate		
6–17	TG	1171 (24.4%)	979 (20.4%)	885 (25.4%)	763 (21.9%)	199 (26.1%)	157 (20.6%)	6.155	0.188
(N = 9039)	TC	1439 (30.0%)	609 (12.7%)	943 (27.1%)	369 (10.6%)	203 (26.6%)	68 (8.9%)	33.37	0.000
	HDL-C	494 (10.3%)	182 (3.8%)	452 (13.0%)	188 (5.4%)	124 (16.3%)	53 (6.9%)	56.848	0.000
	LDL-C	691 (14.4%)	259 (5.4%)	497 (14.3%)	167 (4.8%)	109 (14.3%)	35 (4.6%)	2.092	0.719
6–11	TG	765 (22.8%)	694 (20.7%)	386 (22.0%)	438 (24.9%)	53 (20.6%)	72 (28.0%)	16.789	0.002
(N = 5375)	TC	1106 (33.0%)	452 (13.5%)	544 (31.0%)	202 (11.5%)	78 (30.2%)	32 (12.5%)	9.79	0.044
	HDL-C	254 (7.5%)	90 (2.7%)	156 (8.8%)	76 (4.3%)	30 (11.7%)	17 (6.4%)	25.677	0.000
	LDL-C	518 (15.4%)	177 (5.2%)	474 (15.2%)	86 (4.9%)	40 (15.5%)	13 (4.9%)	0.4	0.982
12–17	TG	406 (28.2%)	285 (19.8%)	499 (29.0%)	325 (18.9%)	146 (28.9%)	85 (16.8%)	2.352	0.671
(N = 3664)	TC	334 (23.2%)	157 (10.9%)	399 (23.2%)	167 (9.7%)	125 (24.8%)	36 (7.1%)	6.733	0.151
	HDL-C	240 (16.7%)	92 (6.4%)	296 (17.2%)	112 (6.5%)	94 (18.7%)	36 (7.1%)	1.645	0.801
	LDL-C	173 (12.1%)	82 (5.7%)	23 (13.5%)	81 (4.7%)	69 (13.7%)	22 (4.4%)	3.854	0.426

**Table 8 nutrients-17-03159-t008:** Critical and abnormal rates of serum lipids among children and adolescents with vitamin D insufficiency and deficiency, stratified by gender.

Age	Blood Lipids (mmol/L)	Normal (N = 4798)		Insufficiency (N = 3479)		Deficiency (N = 762)		χ^2^	*p*
		Critical Rate	Abnormal Rate	Critical Rate	Abnormal Rate	Critical Rate	Abnormal Rate		
6–17	TG	1171 (24.4%)	979 (20.4%)	885 (25.4%)	763 (21.9%)	199 (26.1%)	157 (20.6%)	6.155	0.188
(N = 9039)	TC	1439 (30.0%)	609 (12.7%)	943 (27.1%)	369 (10.6%)	203 (26.6%)	68 (8.9%)	33.37	0.000
	HDL-C	494 (10.3%)	182 (3.8%)	452 (13.0%)	188 (5.4%)	124 (16.3%)	53 (6.9%)	56.848	0.000
	LDL-C	691 (14.4%)	259 (5.4%)	497 (14.3%)	167 (4.8%)	109 (14.3%)	35 (4.6%)	2.092	0.719
Male	TG	680 (24.1%)	541 (19.1%)	387 (24%)	355 (22%)	62 (24.9%)	53 (21.3%)	6.011	0.196
(N = 4687)	TC	808 (28.6%)	353 (12.5%)	429 (26.6%)	168 (10.4%)	60 (24.2%)	26 (10.4%)	11.607	0.021
	HDL-C	301 (10.7%)	113 (4%)	226 (14%)	102 (6.3%)	40 (16.2%)	28 (11.2%)	50.327	0.000
	LDL-C	398 (14.1%)	150 (5.3%)	235 (14.6%)	84 (5.2%)	44 (17.7%)	12 (5%)	2.620	0.623
Female	TG	491 (24.9%)	438 (22.2%)	498 (26.7%)	408 (21.9%)	137 (26.7%)	104 (20.3%)	2.452	0.653
(N = 4352)	TC	631 (32%)	256 (13%)	514 (27.5%)	201 (10.7%)	143 (27.8%)	42 (8.2%)	27.811	0.000
	HDL-C	193 (9.8%)	69 (3.5%)	226 (12.2%)	86 (4.6%)	84 (16.4%)	25 (4.8%)	24.003	0.000
	LDL-C	293 (14.9%)	109 (5.5%)	262 (14.1%)	83 (4.5%)	65 (12.6%)	22 (4.4%)	4.890	0.299

**Table 9 nutrients-17-03159-t009:** Ordinal regression of children and adolescents with critical/abnormal LDL.

Factors	β	Std.E	Wald χ^2^	OR	*p*	95%CI	
						Lower Bound	Upper Bound
Sex	0.037	0.053	0.478	1.04	0.489	0.94	1.15
Age group	−0.122	0.055	4.879	0.89	0.027	0.80	0.99
Location	−0.041	0.053	0.609	0.96	0.435	0.87	1.06
Normal weight	−0.743	0.073	104.238	0.48	0.000	0.41	0.55
Overweight	−0.262	0.091	8.208	0.77	0.004	0.64	0.92
Obesity							
Normal vitamin A	−0.023	0.443	0.003	0.98	0.959	0.41	2.33
Marginal vitamin A deficiency	−0.738	0.464	2.535	0.48	0.111	0.19	1.19
Vitamin A deficiency							
Normal vitamin D	−0.018	0.101	0.031	0.98	0.86	0.81	1.20
Vitamin D Insufficiency	−0.041	0.102	0.164	0.96	0.685	0.79	1.17
Vitamin D Deficiency							

**Table 10 nutrients-17-03159-t010:** Ordinal regression of children and adolescents with critical/abnormal HDL.

Factors	β	Std.E	Wald χ^2^	OR	*p*	95%CI	
						Lower Bound	Upper Bound
Sex	−0.101	0.058	3.02	0.90	0.082	0.81	1.01
Age group	0.892	0.059	229.885	2.44	0.000	2.17	2.74
Location	0.16	0.058	7.625	1.17	0.006	1.05	1.32
Normal weight	−0.943	0.079	143.41	0.39	0.000	0.33	0.45
Overweight	−0.282	0.097	8.553	0.75	0.003	0.62	0.91
Obesity							
Normal vitamin A	−0.734	0.408	3.228	0.48	0.072	0.22	1.07
Marginal vitamin A deficiency	−0.338	0.422	0.643	0.71	0.423	0.31	1.63
Vitamin A deficiency							
Normal vitamin D	−0.386	0.098	15.522	0.68	0.000	0.56	0.82
Vitamin D Insufficiency	−0.211	0.097	4.762	0.81	0.029	0.67	0.98
Vitamin D Deficiency							

**Table 11 nutrients-17-03159-t011:** Ordinal regression of children and adolescents with critical/abnormal TG.

Factors	β	Std.E	Wald χ^2^	OR	*p*	95%CI	
						Lower Bound	Upper Bound
Sex	0.192	0.042	21.146	1.21	0.000	1.12	1.31
Age group	0.021	0.043	0.239	1.02	0.625	0.94	1.11
Location	0.042	0.041	1.023	1.04	0.312	0.96	1.13
Normal weight	−0.813	0.062	169.518	0.44	0.000	0.39	0.50
Overweight	−0.322	0.078	16.896	0.72	0.000	0.62	0.84
Obesity							
Normal vitamin A	0.779	0.41	3.599	2.18	0.058	0.97	4.87
Marginal vitamin A deficiency	−0.298	0.424	0.493	0.74	0.483	0.32	1.70
Vitamin A deficiency							
Normal vitamin D	−0.072	0.078	0.848	0.93	0.357	0.80	1.08
Vitamin D Insufficiency	0.011	0.078	0.021	1.01	0.883	0.87	1.18
Vitamin D Deficiency							

**Table 12 nutrients-17-03159-t012:** Ordinal regression of children and adolescents with critical/abnormal TC.

Factors	β	Std.E	Wald χ^2^	OR	*p*	95%CI	
						Lower Bound	Upper Bound
Sex	0.116	0.042	7.709	1.12	0.005	1.03	1.22
Age group	−0.436	0.044	99.505	0.65	0.000	0.59	0.70
Location	−0.093	0.042	4.936	0.91	0.026	0.84	0.99
Normal weight	−0.388	0.063	37.979	0.68	0.000	0.60	0.77
Overweight	−0.214	0.080	7.201	0.81	0.007	0.69	0.94
Obesity							
Normal vitamin A	0.748	0.407	3.386	2.11	0.066	0.95	4.69
Marginal vitamin A deficiency	−0.008	0.418	0	0.99	0.985	0.44	2.25
Vitamin A deficiency							
Normal vitamin D	0.152	0.081	3.511	1.16	0.061	0.99	1.36
Vitamin D Insufficiency	0.011	0.082	0.018	1.01	0.894	0.86	1.19
Vitamin D Deficiency				1.00			

## Data Availability

Supporting data can be acquired from the corresponding author.

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
