# Peer review of "Associations Between Lipid Profiles and Vitamin A and D Deficiencies Among Children and Adolescents in Zhejiang Province, China: A Cross-Sectional Study"

_nutrients, 2025, doi:10.3390/nu17193159_

Round 1
Reviewer 1 Report
Comments and Suggestions for Authors
Dear Authors, I have read your manuscript that evaluated the association between lipid profiles and vitamin A and vitamin D deficiencies in children and adolescents. I send you my comments.
1) Methods: please clarify the type of study; I think that it is an interventional study because you take a blood sample.
2) Methods: please add where the patients were referred
3) Methods:Statistics: please add the power calculation; please add the statistical evaluation using the student t-test and the Mantova test.
4) Tables: these are very hard to understand. Please revise to show data for males and females in the urban area and males and females in the urban area. Then add a table showing only a statistical evaluation between these areas.
5) Table 2: Please revise the character; I think that it is a Chinese language.
6) Table 2, 3, and 4: please report the absolute data, not the percentage.
7) For Tables 2, 3, and 4, please add the differentiation with respect to males and females.
8) Results: please add the family data; usually metabolic diseases show a familiar or a genetic pattern, therefore could you exclude these roles? If you think that the lifestyle as well as the diet plays a role, please add these data into the results.
9) The title of the manuscript is “Associations Between Lipid Profiles and Vitamin A and D Deficiencies ….” So why you describe into the discussion the obesity ? You need to add biochemical data evaluating the mechanism of this correlation, and you need to explain which measure could be taken into account to improve these data that could induce the development of severe systemic diseases.
10) Discussion: I have not read any data with respect to vitamin A.
11) Discussion: It seems to be a descriptive report of the results; please rewrite it.
12) Text: please change "gender" to "sex."
Author Response
Dear Editors and Reviewers:
Thank you for your comments concerning our manuscript entitled “Associations between lipid profiles and vitamin A and D deficiencies among children and adolescents in Zhejiang Province, China: A cross-sectional study ( nutrients-3907126 ). The comments are all valuable and very helpful for revising and improving our paper, as well as the important guiding significance to our researches. We have studied comments carefully and have made correction which we hope to meet with approval. The main corrections in the paper and the responds are as flowing:
Reviewer1
- Methods: please clarify the type of study; I think that it is an interventional study because you take a blood sample.
Response: Our study is a cross-sectional study, not an interventional study. The blood samples were collected solely for the purpose of measuring lipid profiles and vitamin levels, without any intervention administered to the participants. We have added a clarification sentence at the beginning of the “Methods” section to emphasize this point.
The cross-sectional study was conducted from 2022 to 2024 and included children and adolescents aged 6-17 years who participated in the provincial nutrition surveillance.
Blood samples were collected from the participants to measure the concentrations of triglycerides (TG), total cholesterol (TC), high-density lipoprotein cholesterol (HDL-C), low-density lipoprotein cholesterol (LDL-C), vitamin A (VA), and vitamin D (VD), and no interventions were administered to the participants.
- Methods: please add where the patients were referred
Response: We have added the relevant information to the “Methods” section to specify the recruitment sources of the participants.
The participants were recruited from 90 counties (cities and districts) across Zhejiang Province as part of the provincial nutrition surveillance program, which aims to assess the nutritional status of the pediatric population in the region.
- Methods: Statistics: please add the power calculation; please add the statistical evaluation using the student t-test and the Mantova test.
Response: We have made the following revisions to address your concerns:
Power Calculation: We have added a detailed power calculation section to our Methods section. The sample size was determined based on the prevalence of dyslipidemia in previous studies. Assuming a prevalence of 20%, a power of 80%, and a significance level of 0.05, the required sample size was calculated to be 1537 participants per group. Given that our study included both urban and rural children and adolescents, and considering the need for adequate representation from both groups, the total sample size was adjusted accordingly. The final sample size of 9039 participants, accumulated over three years of monitoring (2022-2024), ensures sufficient statistical power to detect significant differences in lipid profiles across various subgroups. This comprehensive sample size allows for robust analysis and meaningful conclusions regarding the prevalence and determinants of dyslipidemia in the pediatric population of Zhejiang Province.
Statistical Evaluation: We have expanded our statistical analysis to include Student's t-tests for normally distributed continuous variables and multivariate analysis of variance (MANOVA) to assess the overall differences in lipid profiles across different groups.
2.6. Statistical Analysis
Median, 25% and 75% percentiles are presented for the continuous variables. Comparisons were performed using Mann-Whitney U nonparametric test for the continuous variables and chi-squared test for categorical variables. To further evaluate the differences in means for normally distributed continuous variables, Student's t-tests were conducted. Additionally, multivariate analysis of variance (MANOVA) was performed to assess the overall differences in lipid profiles across different groups. Ordinal regression models were employed to explore the influencing factors of critical/abnormal TG, TC, HDL-C, LDL-C. Data processing and statistical analyses were performed using SAS 9.2 software (SAS Institute, Cary, NC, USA). All tests were two-sided, and the level of significance was set at P < 0.05.
Power Calculation: The sample size for this study was determined based on the prevalence of dyslipidemia in previous studies. Assuming a prevalence of 20%, a power of 80%, and a significance level of 0.05, the required sample size was calculated to be 1537 participants per group. Given that our study included both urban and rural children and adolescents, and considering the need for adequate representation from both groups, the total sample size was adjusted accordingly. The final sample size of 9039 participants, accumulated over three years of monitoring (2022-2024), ensures sufficient statistical power to detect significant differences in lipid profiles across various subgroups. This comprehensive sample size allows for robust analysis and meaningful conclusions regarding the prevalence and determinants of dyslipidemia in the pediatric population of Zhejiang Province.
- Tables: these are very hard to understand. Please revise to show data for males and females in the urban area and males and females in the urban area. Then add a table showing only a statistical evaluation between these areas.
Response:We have revised the tables to better present the data and facilitate easier comparison between different groups. Specifically, we have restructured Table 1 to separately show the serum lipid concentrations for males and females in both urban and rural areas. Additionally, we have created a new Table 2 that exclusively presents the statistical evaluations.
- Table 2: Please revise the character; I think that it is a Chinese language.
Response: We have carefully reviewed the table and have made the revisions to ensure that all characters and descriptions are in English.
- Table 2, 3, and 4: please report the absolute data, not the percentage. For Tables 2, 3, and 4, please add the differentiation with respect to males and females.
Response: We have made the following revisions to address your concerns:
Absolute Data Reporting: We have updated Tables 2, 3, and 4 to report the absolute counts instead of percentages.
Differentiation by Sex: We have added differentiation by sex for Tables 3-8. The data are now presented separately for males and females.
- Results: please add the family data; usually metabolic diseases show a familiar or a genetic pattern, therefore could you exclude these roles? If you think that the lifestyle as well as the diet plays a role, please add these data into the results.
Response:We appreciate your suggestion regarding the inclusion of family data and dietary habits. Unfortunately, we did not collect these data in the current study. However, we have acknowledged this limitation in the discussion section of our manuscript. We have added a statement to highlight the importance of considering family history and detailed dietary habits in future research. We plan to include these variables in our next study to better understand their impact on lipid profiles and vitamin status.
We have revised the discussion section to reflect these limitations as follows:
Additionally, we did not collect data on family history of metabolic diseases or detailed dietary habits, such as eating out, picky eating, or food preferences. These factors may play a role in lipid profiles and vitamin status. In future studies, we plan to include these variables to better understand their impact. Although we analyzed the relationship between individual physical activity levels and lipid indicators and found no association, it is still important to consider the potential influence of family history and diet in future research.
- The title of the manuscript is “Associations Between Lipid Profiles and Vitamin A and D Deficiencies ….” So why you describe into the discussion the obesity ? You need to add biochemical data evaluating the mechanism of this correlation, and you need to explain which measure could be taken into account to improve these data that could induce the development of severe systemic diseases.
Response:We have revised the “Discussion” section to include a more detailed explanation of the biochemical mechanisms underlying the association between obesity and lipid abnormalities. Specifically, we have added a discussion on how obesity and overweight can lead to insulin resistance, which in turn affects lipid metabolism. Insulin resistance often results in increased hepatic production of very low-density lipoprotein (VLDL), leading to elevated TG levels. Additionally, it impairs the clearance of TG-rich lipoproteins, further exacerbating hypertriglyceridemia[26]. This provides a clearer understanding of the metabolic pathways involved in the development of dyslipidemia in obese and overweight children and adolescents. Furthermore, we have included recommendations for public health initiatives. We emphasize the importance of promoting a balanced diet rich in fruits, vegetables, whole grains, and healthy fats, while reducing the intake of saturated and trans fats. Regular physical activity is also recommended to improve overall metabolic health and reduce the risk of obesity-related complications[27].
- Discussion: I have not read any data with respect to vitamin A.
Response:We have included a new subsection titled “4.5. Vitamin A Status and Lipid Profiles” where we discuss the significant associations observed in our study between vitamin A status and lipid profiles among children and adolescents.
4.5. Vitamin A Status
In our study, we observed significant associations between vitamin A status and lipid profiles in the univariate analysis. Vitamin A deficiency was associated with higher critical and abnormal rates of TC and LDL-C. These findings suggest that maintaining adequate levels of vitamin A may be important for healthy lipid profiles in children and adolescents.Vitamin A deficiency has been associated with alterations in lipid profiles, possibly through its effects on liver function and lipid transport mechanisms[28]. However, these associations did not reach statistical significance in the ordinal regression analysis, possibly due to the adjustment for multiple confounding variables, small sample sizes in certain subgroups, and the complex interplay of other unmeasured factors.
Discussion: It seems to be a descriptive report of the results; please rewrite it.
Response: We have revised the “Discussion” to provide a more comprehensive and insightful analysis of our results, exploring the potential mechanisms, implications, and future research directions.
Key Revisions:
4.2. Gender Differences
Our study revealed significant gender and urban-rural disparities in serum lipid levels. Boys had higher mean serum TG levels, while girls had higher mean serum TC levels. Urban children had higher mean serum TC, HDL-C, and LDL-C levels compared to their rural counterparts.
4.3. Urban-Rural Disparities
Obesity and overweight status were significantly associated with elevated TG, TC, low HDL-C, and elevated LDL-C. Children and adolescents with obesity or overweight had higher prevalence rates of dyslipidemia compared to their normal-weight counterparts.
4.5. Vitamin A Status
In our study, we observed significant associations between vitamin A status and lipid profiles in the univariate analysis. Vitamin A deficiency was associated with higher critical and abnormal rates of TC and LDL-C. These findings suggest that maintaining adequate levels of vitamin A may be important for healthy lipid profiles in children and adolescents.Vitamin A deficiency has been associated with alterations in lipid profiles, possibly through its effects on liver function and lipid transport mechanisms[28]. However, these associations did not reach statistical significance in the ordinal regression analysis, possibly due to the adjustment for multiple confounding variables, small sample sizes in certain subgroups, and the complex interplay of other unmeasured factors.
4.6. Vitamin D Status
In contrast, vitamin D deficiency was significantly associated with elevated TC and low HDL-C in the ordinal regression analysis. This finding underscores the importance of vitamin D in lipid metabolism. Vitamin D plays a critical role in calcium metabolism, bone health, and immune regulation. Recent evidence suggests that vitamin D plays a role in regulating lipid metabolism, and its deficiency may contribute to adverse lipid profiles[29]. The mechanisms underlying this association are not fully understood, but vitamin D is known to influence lipid metabolism through its effects on the parathyroid hormone (PTH) and calcium levels, which in turn affect lipid synthesis and transport[30].
4.7. Mechanisms and Future Research
The mechanisms underlying the associations between vitamin A and D deficiencies and lipid profiles are not fully understood. Vitamin A is essential for the proper functioning of the retinoid X receptor (RXR), which is involved in the regulation of lipid homeostasis[31]. Deficiency in vitamin A may impair the activity of RXR, leading to dysregulation of lipid synthesis and transport. Vitamin D also plays a role in lipid metabolism, and its deficiency may contribute to adverse lipid profiles. Future studies should explore the underlying mechanisms and the potential benefits of vitamin supplementation in improving lipid profiles in children and adolescents.
- Text: please change "gender" to "sex."
Response: Thank you for your comment regarding the terminology used in the manuscript. We have revised the text to replace "gender" with "sex" throughout the document to more accurately reflect biological differences.
We are very sorry for our negligence in preparing the manuscript and we have made corrections on the manuscript. We appreciate for your warm work earnestly, and hope that the correction will meet with approval. Once again, thank you very much for your comments and suggestions.
Sincerely,
Yan Zou
Reviewer 2 Report
Comments and Suggestions for Authors
I have read this paper with interest, and value the effort. It is hereby clearly reflected that the authors respect the exploratory study design, and therefore report their results with association type of language, not necessary reflecting causality.
The observation on gender (e.g. 4.2.) is an interesting one, while your cohort including both pre-pubertal to mature adolescents, like Tanner criteria or similar. Do you have any data to explore this association, or is there eg an age-dependent effect of gender ? Along the same line, was the incidence of obesity similar in both genders ?
Are there ‘nutritional’ or other strategies that could attain both targets simultaneously (reduce elevated TC and low HDL-C and improve vit D deficiency). Related to this, why are there specific concerns for the Zhejiang Province.
On the sampling, was there a strategy of the timing of food intake to sampling?
Can the authors reflect on the ‘appropriateness’ of the cohort to the population ‘targeted’.
I support the current statistical analysis as described and performed by the authors.
Author Response
Dear Editors and Reviewers:
Thank you for your comments concerning our manuscript entitled “Associations between lipid profiles and vitamin A and D deficiencies among children and adolescents in Zhejiang Province, China: A cross-sectional study ( nutrients-3907126 ). The comments are all valuable and very helpful for revising and improving our paper, as well as the important guiding significance to our researches. We have studied comments carefully and have made correction which we hope to meet with approval. The main corrections in the paper and the responds are as flowing:
Review2
I have read this paper with interest, and value the effort. It is hereby clearly reflected that the authors respect the exploratory study design, and therefore report their results with association type of language, not necessary reflecting causality.
1.The observation on gender (e.g. 4.2.) is an interesting one, while your cohort including both pre-pubertal to mature adolescents, like Tanner criteria or similar. Do you have any data to explore this association, or is there eg an age-dependent effect of gender ? Along the same line, was the incidence of obesity similar in both genders ?
Response: Unfortunately, our dataset does not include Tanner stage information, which limits our ability to explore age-dependent gender effects on lipid profiles. We acknowledge this as a limitation and suggest future studies incorporate Tanner stages to better understand these associations. Our analysis shows that the prevalence of obesity and overweight status differs significantly between boys and girls (P<0.05), with boys having a higher incidence. This finding highlights the need for gender-specific considerations in public health interventions targeting obesity and lipid profiles. We have revised the discussion section to enhance the clarity and scientific rigor of our manuscript.
Our study revealed significant sex and urban-rural disparities in serum lipid levels. Boys had higher mean serum TG levels, while girls had higher mean serum TC levels. Urban children had higher mean serum TC, HDL-C, and LDL-C levels compared to their rural counterparts. These findings are in line with previous research that has identified sex differences in serum lipid levels [20].Although our dataset did not include Tanner stages to further investigate the interaction between sex and pubertal development, we recognize the importance of this factor. Future studies should consider incorporating Tanner stages to elucidate the relationship between gender, puberty, and lipid profiles. The observed gender differences may be attributed to variations in lifestyle, dietary habits, and genetic predispositions. Future interventions should account for these differences to develop effective public health strategies.
The metabolic interlink between hypertriglyceridemia and low HDL-C levels is strongly associated with cardiovascular disease (CVD) in both adults and children[25]. Our analysis also found that the prevalence of obesity and overweight was significantly different between boys and girls and this may be related to differences in growth patterns and lifestyle habits during the peripubertal period.
- Are there ‘nutritional’ or other strategies that could attain both targets simultaneously (reduce elevated TC and low HDL-C and improve vit D deficiency). Related to this, why are there specific concerns for the Zhejiang Province.
Response: Addressing both elevated TC, low HDL-C, and vitamin D deficiency simultaneously is a multifaceted challenge. Dietary interventions that emphasize a balanced intake of fatty acids, including increasing consumption of omega-3 fatty acids and reducing saturated and trans fats, could help improve lipid profiles. Vitamin D supplementation, particularly in the context of improved sun exposure and fortified foods, may address vitamin D deficiency while also having beneficial effects on lipid metabolism. Future interventions should consider integrated approaches that target multiple aspects of metabolic health.The specific concerns for Zhejiang Province arise from its unique demographic and lifestyle characteristics. The province's rapid urbanization and economic development have led to significant changes in diet and physical activity levels, contributing to a higher prevalence of obesity and related metabolic disorders.
We have revised the manuscript to better address these points and to provide a more detailed discussion on potential strategies and the specific context of Zhejiang Province.
- On the sampling, was there a strategy of the timing of food intake to sampling?
Response: In our study, all blood samples were collected after an overnight fast of at least 8 hours. Participants were explicitly instructed to refrain from eating or drinking anything except water for the 8 hours preceding the blood draw. This standard fasting protocol is widely used in clinical and epidemiological studies to ensure that lipid measurements are not affected by recent food intake. Therefore, we are confident that our lipid measurements reflect the fasting state and are not influenced by the timing of food intake. We have clarified this in the Methods section of our manuscript to ensure transparency.
2.3. Laboratory Assessment
Blood samples were collected after an overnight fast of at least 8 hours. Participants were explicitly instructed to refrain from eating or drinking anything except water for the 8 hours preceding the blood draw. This standard fasting protocol ensures that lipid measurements are not affected by recent food intake.
4.Can the authors reflect on the ‘appropriateness’ of the cohort to the population ‘targeted’.
I support the current statistical analysis as described and performed by the authors.
Response:Our study involved three independent cross-sectional surveys conducted in 2022, 2023, and 2024, respectively. Each survey captured a representative sample of children and adolescents in Zhejiang Province. This design allows us to assess the prevalence and associations of lipid profiles and vitamin deficiencies at different time points. Combining the data enhances the robustness of our findings and provides a comprehensive understanding of the current health landscape in the province. While cross-sectional studies cannot establish causality, they are suitable for identifying prevalence and associations within a population at a specific point in time.
We have added a brief reflection on the appropriateness of our study design in the Discussion section to address this point.
4.8. Strengths and Limitations
4.8.1. Strengths
Our study included three independent cross-sectional surveys conducted in 2022, 2023, and 2024, each capturing a representative sample of children and adolescents in Zhejiang Province. This approach allows us to assess the prevalence and associations of lipid profiles and vitamin deficiencies at different time points, enhancing the robustness of our findings and providing a comprehensive understanding of the current health landscape in the province. While cross-sectional studies cannot establish causality, they are well-suited for identifying prevalence and associations within a population at a specific point in time.
We are very sorry for our negligence in preparing the manuscript and we have made corrections on the manuscript. We appreciate for your warm work earnestly, and hope that the correction will meet with approval. Once again, thank you very much for your comments and suggestions.
Sincerely,
Yan Zou
Round 2
Reviewer 1 Report
Comments and Suggestions for Authors
in my opinion the manuscript has been improved